# Non-Fungible Tokens Based on ERC-4519 for the Rental of Smart Homes

**DOI:** 10.3390/s23167101

**Published:** 2023-08-11

**Authors:** Javier Arcenegui, Rosario Arjona, Iluminada Baturone

**Affiliations:** Instituto de Microelectrónica de Sevilla (IMSE-CNM), Universidad de Sevilla-CSIC, C/Américo Vespucio 28, 41092 Seville, Spain; arcenegui@imse-cnm.csic.es (J.A.); arjona@imse-cnm.csic.es (R.A.)

**Keywords:** blockchain, smart home, Internet of Things, IoT, smart contract, real estate, non-fungible token, NFT, Ethereum

## Abstract

The rental of houses is a common economic activity. However, there are many inconveniences that arise when renting a property. The lack of trust between the landlord and the tenant due to fraud or squatters makes it necessary to involve third parties to minimize risk. A blockchain (such as Ethereum) provides an ideal solution to act as a low-cost intermediary. This paper proposes the use of non-fungible tokens (NFTs) based on ERC-4519 for smart home tokenization. The ERC-4519 is an Ethereum standard for describing NFTs tied to physical assets, allowing smart homes (assets) to be linked to NFTs so that the smart homes can interact with the blockchain and perform transactions, know their landlord (owner) and assigned tenant (user), whether they are authenticated or not, and know their operating mode (NFT state). The payments associated with the rental process are made using the NFT, eliminating the need for additional fungible tokens and simplifying the process. The entire rental process is described and illustrated with a proof of concept using a Pycom Wipy 3.0 as a smart home gateway and a smart contract programmed in Solidity, which is deployed on the Goerli Testnet for Ethereum. Experimental results show that the smart home gateway takes a few tens of milliseconds to complete a transaction, and the transaction costs of the relevant functions of the smart contract are quite affordable.

## 1. Introduction

A major problem in the current state of rental markets is the lack of reliability in payments on the side of the tenant and the veracity of the property on the side of the landlord. This has led to the use of third-party intermediaries, such as real estate agencies, to avoid such issues. However, this approach often involves high costs and does not fully address the situation. Blockchain technologies are increasingly being used in this area to act as a low-cost intermediary between the landlord and the tenant [1,2,3]. Since a blockchain is a distributed and cryptographically secure chain of data blocks, it is assumed that the information written on the blockchain is immutable. Hence, transparency and traceability are enhanced in the rental process, as both the tenant and the landlord can track the state of the home and confirm whether it is truly rented or not and how.

Among the available blockchains, this paper focuses on Ethereum, which is one of the largest public blockchains in the world capable of executing smart contracts. The use of a public blockchain for home rentals is interesting because it makes it easy for anyone to act as a tenant or landlord. Smart contracts, which are deterministic programs that run on the blockchain, offer many advantages for replacing traditional rental contracts [4].

In addition, due to the evolution of home technology and the growth of the Internet of Things (IoT), many of today’s homes are referred to as smart homes because of their ability to make the lives of their occupants easier by minimizing the amount of effort required to maintain the home or its occupants’ well-being. Smart homes are equipped with a set of sensors and actuators that are often connected to a central data computing system, referred to as a gateway, which serves to transmit and receive information from the Internet [5]. The gateway can be in charge of controlling simple tasks such as adjusting the temperature or lighting of a room, as well as more complex ones that involve communication with external entities that provide services. In this paper, we propose connecting the smart home gateway to Ethereum. As already shown in [6,7], the integration of smart homes with blockchain technology enables the management of home access control, identification/authentication of its occupants, and secure data transmission, which are interesting advantages in the field of smart home rentals. In this paper, we propose the idea that the smart home plays an active role in the rental process, as well as the landlord and the tenant. As described below, the proposal is to represent the smart home with a token in Ethereum.

Ethereum, as a public blockchain, allows the community to improve its functionality through Ethereum Improvement Proposals (EIPs). There are six types of EIPs with different purposes. The ERC (Ethereum Request for Comments) types are used to create and manage tokens in a secure way. A token is a digital asset that is owned by a blockchain participant. A token can be fungible or non-fungible. Fungible tokens are like casino tokens. They are all the same and have the same specific value. In Ethereum, the ERC-20 defines the most widely used standard for creating fungible tokens [8]. Tokenization has been used for real estate assets to make them liquid, secure, and efficient. Since the initial cost required to buy a property is very high, making real estate unattractive to the retail investor, fungible tokens have been used to represent small percentages of properties, allowing many investors to purchase tokens that represent incredibly small percentages of the underlying assets [9]. Instead of dividing a smart home into fungible tokens, in this paper, we focus on representing the smart home with a non-fungible token (NFT). NFTs are unique tokens associated with a physical or digital asset that could have a different value. In Ethereum, the ERC-721 defines the most widely used standard for creating non-fungible tokens [10]. ERC-721 mainly defines the token identifier and the address of its owner. Since Ethereum uses public-key cryptography, the address of a participant is associated with the public key, which allows for the verification of the digital signatures of that participant.

The ERC-4519 [11] is an extension of the ERC-721, allowing a smart asset to be tied to an NFT. The NFT contains the Ethereum address of the asset as one of the attributes, and the asset is the only one able to generate that address [12]. This way, the home’s authenticity can be checked as a way to prevent fraud. As a second advantage, an ERC-4519 NFT contains the Ethereum address of the asset’s user, as well as the owner’s address. This prevents squatters from being users of the home. As a third advantage, an ERC-4519 NFT allows for online authentication between the owner and the token, as well as between the user and the token [13]. Hence, the impersonation of users, owners, and the homes themselves is avoided. All these advantages make ERC-4519 an ideal choice for tokenizing a home for rental purposes.

This work presents the following contributions to the field of blockchain-powered smart homes:The linking of a smart home to an NFT using the smart home gateway as a smart asset. Since a gateway has enough computing power, it can securely generate its own blockchain account. By storing the address of this account in the token, the smart home can authenticate itself by signing with the private key associated with the blockchain account.The mutual authentication between the landlord (owner) and the smart home (asset) on one side, and between the tenant (user) and the smart home (asset) on the other side. This is possible because the token has defined not only its owner but also its users. This ensures secure communication between the parties, preventing unauthorized access to the house.The operating modes of the smart home gateway are linked to the states of the NFT. Hence, the smart home knows at all times whether the landlord and tenant are authenticated or not and operates accordingly.The same NFT is used to make the payments associated with the rental process. This approach simplifies the process by eliminating the need for security tokens to make the payments.A proof of concept of the whole rental process is presented, which uses a Pycom Wipy 3.0 as the smart home gateway and a smart contract programmed in Solidity and deployed on the Goerli Testnet, which is a testing network for Ethereum.

The rest of this paper is structured as follows: Section 2 reviews the related works on using blockchains (particularly Ethereum) for the rental of smart homes. Section 3 summarizes the main features of ERC-4519 NFTs. Section 4 presents information on how to use the ERC-4519 for smart home rentals. Section 5 describes the realization of the proof of concept and the experimental results obtained. Finally, Section 6 presents the conclusions.

## 2. Related Works

In the literature, there are a few proposals for renting smart homes based on blockchains. The work in [1] proposes a solution to convert traditional rental agreements into smart contracts. The functions implemented in the smart contract are contract activation and signatures, deposit payments, water/electricity meter recordings, rent payments, completion of the rent process, refund of the deposit amount, acceptance/rejection of the deposit amount refund, and retrieval of the remaining rent balance. Similar functions are considered in the proposal in [2]. The main goal of these two works is to make the rental process secure, traceable, and visible, without the need for a central authority or intermediaries. Other works, such as [3], maintain the need for a trusted authority, but only for the arbitration and tracing of malicious users and to improve the feasibility of the solution with a reputation mechanism. In these works, the house is a passive actor in the process.

The work in [4] focuses on the buying and selling of smart homes in smart cities. If an individual requests access to a property, the house, acting as an IoT, receives the request, reads the blockchain, and approves access if the individual is the owner, or denies it otherwise. The work in [6] provides a solution for renting smart homes that includes the registration of an IoT device included in the smart home using the manufacturer’s blockchain address and the device information (e.g., the electronic product code). The IoT device (for example, an IP camera) represents the smart home. The solution is based on a smart contract that defines the functionalities of checking the ownership information, IoT device authentication, ownership transfer, and tenancy transfer. The smart contract is created by the manufacturer of the IoT device.

All the solutions referred to above are based on smart contracts. However, none of them employ home tokenization. The work in [9] introduces the tokenization of homes as a way to increase the security and liquidity of the real estate market and reduce the administrative burden and costs involved in buying and selling properties. Every token holder has some percentage of ownership in the home. The authors employ fungible tokens based on the ERC-777 token standard, which improves the widely used ERC-20 token standard to offer token holders more control over their tokens [14]. In the proposal in [15], each smart home has a gateway that manages and controls access to IoT devices by external services (such as garbage collection) according to smart contracts. They use fungible tokens for the payments of services but do not use tokens to represent the home.

None of the above-mentioned works use NFTs. The most widely used token standard to define NFTs in Ethereum is the ERC-721 [10]. It describes the basic attributes that an NFT should possess, including its identifier and owner (and who can manage the NFTs of an owner), and provides the basic functions to track and transfer the ownership of NFTs (which can represent digital or physical assets). The work in [16] proposes a system for exchanging or selling real estate assets as ERC-721 NFTs.

Table 1 summarizes the main features of the above-mentioned works on blockchains and smart homes.

Concerning tokens in Ethereum, the ERC-1155 (Multi-Token Standard) [17] was proposed to allow for any combination of fungible tokens, non-fungible tokens, or other configurations (e.g., semi-fungible tokens). A limitation of ERC-721 NFTs and the ERC-1155 is that they do not consider users. Among the ERCs that consider users, the ERC-4494 (Permit for ERC-721 NFTs) [18] and the ERC-5334 (ERC-721 User And Expires And Level Extension) [19] are in the “Draft” state, which is the initial state of any EIP, so they may suffer variations before entering the “Final” state if they reach such a state.

Among the ERCs dedicated to rental NFTs, only the ERC-4907 (Rental NFT, an extension of ERC-721) is currently in the “Final” state [20]. It adds the user role and an expiration time for the use of the NFT. The ERC-5187 (Extend ERC-1155 with rentable usage rights) [21] and the ERC-5501 (Rental and Delegation NFT—ERC-721 Extension) [22] are currently in the “Draft” state. The ERC-2615 (Non-Fungible Token with mortgage and rental functions) is currently in the “Stagnant” state, which means that its authors did not solve the flaws or incorporate the suggestions for its improvement [23].

Table 2 shows the features of the above-mentioned ERCs and the ERC-4519. The ERC-4519 (Non-Fungible Tokens Tied to Physical Assets) [11], currently in the “Final” state, is the only standard that defines how to tie a physical asset (in this case, the smart home) to the NFT smart contract using a blockchain address. Since this tie is performed through the blockchain address, the asset can interact with the NFT smart contract by signing messages and transactions. Another advantage of the ERC-4519 is that it considers operating modes and allows for the establishment of secure communication channels between the physical asset, its owner, and its user. The following sections describe the ERC-4519 and its use for renting smart homes.

## 3. ERC-4519: Non-Fungible Tokens Tied to Physical Assets

The ERC-4519 [11] consists of attributes, events, and functions that extend the ERC-721 [10]. The attributes of the ERC-721 include *tokenId*, *owner*, and *approved*. The attributes of the ERC-4519 can be described as follows: *tokenId* is a numeric value that identifies the NFT; *owner* is a blockchain address that identifies the owner of the NFT; *asset* is a blockchain address that identifies the physical asset tied to the NFT; *user* is a blockchain address that identifies the user of the NFT; *approved* is a blockchain address that indicates who can transfer the NFT; *state* is a numeric value associated with the NFT states (“*waitingForOwner*”, “*engagedWithOwner*”, “*waitingForUser*”, and “e*ngagedWithUser*”) that indicates whether the *owner*/*user* and the *asset* are authenticated to each other; *hashK_OA* is the hash value of the shared key between the owner and the asset; *hashK_UA* is the hash value of the shared key between the user and the asset; *dataEngagement* is a numeric value containing temporary data for the authentication process; *timestamp* is a numeric value containing the last time the asset executed the smart contract; and *timeout* is a numeric value containing the maximum time set for two executions of the smart contract by the asset (after this time, the asset is considered out of service). Figure 1 shows the flowchart of the NFT states, with the *asset* and *user* addresses defined.

Regarding ERC-4519 events, the *UserAssigned* event is emitted when the NFT is assigned to a new *user*, the *UserEngaged* event is emitted when the *user* and the *asset* successfully complete the mutual authentication process, the *OwnerEngaged* event is emitted when the *owner* and the *asset* successfully complete the mutual authentication process, and the *TimeoutAlarm* event is emitted when the *timestamp* of the NFT is not updated within the *timeout*. None of these events are included in the ERC-721.

The ERC-4519 functions are as follows: *setUser* (which is executed by the *owner*) defines the new *user* of the NFT and changes its state to “*waitingForUser*”; *startOwnerEngagement* (which is executed by the *owner*) defines the initialization of the mutual authentication process between the *owner* and the *asset*; *ownerEngagement* (which is executed by the *asset*) completes the mutual authentication process between the *owner* and the *asset* if *hashK_OA* matches *hashK_A* (hash of the secret generated by the *asset* to share with the *owner*), changes the NFT state to “*engagedWithOwner*”, and emits the *OwnerEngaged* event; *startUserEngagement* (which is executed by the *user*) defines the initialization of the mutual authentication process between the *user* and the *asset*; *userEngagement* (which is executed by the *asset*) completes the mutual authentication process between the *user* and the *asset* if *hashK_UA* matches *hashK_A* (hash of the secret generated by the *asset* to share with the *user*), changes the NFT state to “*engagedWithUser*”, and emits the *UserEngaged* event; *checkTimeout* (which can be executed by everybody) checks whether the *timeout* has expired and emits the *TimeoutAlarm* event; *setTimeout* (which is executed by the *owner*) sets the value of the *timeout* attribute; *updateTimestamp* (which is executed by the *asset*) updates the *timestamp* attribute, thus avoiding the *timeout* alarm; *tokenFromBCA* (which can be executed by anyone) enables the retrieval of the attribute values of the *tokenId* from an address; *ownerOfFromBCA* (which can be executed by anyone) allows for the determination of the *owner* of the token from the address of the *asset* tied to the token; *userOf* (which can be executed by anyone) allows for the determination of the *user* of the token from the *tokenId* attribute; *userOfFromBCA* (which can be executed by anyone) allows for the determination of the *user* of the token from the *asset* attribute (address of the *asset* tied to the token); *userBalanceOf* (which can be executed by anyone) allows for the determination of the number of tokens assigned to a *user*; and *userBalanceOfAnOwner* (which can be executed by anyone) allows for the determination of the number of tokens of a particular *owner* assigned to a *user*. None of these functions are included in the ERC-721.

Depending on the application, some or all of the attributes, events, and functions described above may be used. For applications that do not require a tie between the physical asset and the NFT, only the functionalities associated with the *tokenId*, *owner*, and *user* attributes are considered. This is the case, for example, in the lending of books, unique game cards, or clothing. For applications that do not require users, the functionalities associated with the *user* and *hashK_UA* attributes are not considered. This is the case, for example, in the buying or selling of properties. For applications that require users and a tie between the physical asset and the NFT, all the functionalities are considered. This is the case, for example, in the rental of high-valued assets or assets shared by multiple users. This work focuses on the latter case for a use case of renting smart homes. In the following section, an ERC-4519-based interface is defined for the smart home rental application. The additional attributes, events, and functions required by this application are also described.

## 4. Using the ERC-4519 for Smart Home Rentals

A smart contract for renting smart homes was developed that incorporates the ERC-4519 interface (and also employs the ERC-165, Standard Interface Detection, to identify the interface). All the attributes, events, and functions employed in this application are defined in the following.

### 4.1. Definition of Attributes

Table 3 summarizes the ERC-4519 attributes considered for the rental housing application. The gray attributes on the left are typical of ERC-4519 NFTs, whereas those on the right are typical of smart homes represented by ERC-4519 NFTs.

A smart home is defined by an Internet-of-Things (IoT) gateway and end devices (such as sensors that measure water, diesel, gas, or electricity consumption). The IoT gateway is in charge of facilitating communication between the end devices, both amongst themselves and with the cloud. It also manages data to/from the end devices and provides security to the smart home by mitigating risks. Since the IoT gateway is the central device of the smart home, it is considered an asset tied to the ERC-4519 NFT.

The rental process requires a landlord, who is the owner of the smart home, and a tenant, who is the user of the smart home. Therefore, the tenant and the landlord are considered to be the *owner* and *user* of the ERC-4519 NFT, respectively. If there are multiple owners of a smart home, the owner can be the Ethereum address of a smart contract that defines the owners of the smart home.

Also, the process of renting a smart home requires a payment agreement for the temporary use of the property. Typically, the tenants have to pay an initial deposit to guarantee that they will pay any potential expenses and damages, a rental price (which may include regular charges incurred by the owner), and the water, diesel, gas, or electricity charges. To take this into account, the ERC-4519 should include the following additional attributes: *deposit* and *rentalPrice*, as well as *waterMeter*, *dieselMeter*, *gasMeter*, and *electricityMeter* to register, respectively, the water, diesel, gas, and electricity consumption. If some of these attributes are not needed, they can be set to 0. The temporary use is established by the *rentalTime* attribute. The price of each expenditure is determined by the *waterPrice*, *dieselPrice*, *gasPrice*, and *electricityPrice* attributes.

Finally, a smart home has the ability to check its status and determine whether the tenant is causing damage. These issues must be considered by the landlord to determine the cost through the *homeIssues* attribute. If necessary, the landlord will deduct the cost from the security deposit.

### 4.2. Definition of Events and Functions

In this work, it is assumed that the smart home is equipped with a gateway and sensors and is owned by a landlord. Prior to the rental process, it is also assumed that an ERC-4519 NFT smart contract has been developed, with the *asset* attribute associated with the Ethereum address of the smart home gateway and the *owner* attribute associated with the Ethereum address of the landlord.

The landlord can modify the price of any expenses by executing the *setExpensesPrices* function, which sets the prices of water, diesel, gas, and electricity (0 if the service is not needed). Then, the *Expenses_Prices* attribute (a struct with all the prices) is modified. These prices can be determined by executing the function *getExpensesPrices*, which can be executed by anyone.

If the smart home gateway detects an issue during the rental period, it executes the *newHomeIssue* function. This function updates the *homeIssues* attribute to register the new issue and the *timestamp* attribute and sends the *HomeIssue* event to the landlord to notify that a new issue has been produced.

The functions and events provided in the smart contract are shown in Table 4. The functions depicted in gray (in the upper rows) are typical of ERC-4519 NFTs, whereas those in black are typical of smart homes represented by ERC-4519 NFTs. The following subsections describe the rental setup, rental renewal, and rental termination processes using these functions and events.

#### 4.2.1. Rental Setup

Upon agreement between the tenant and the landlord regarding the rental price and period, the following steps are carried out:The landlord (owner of the ERC-4519 NFT) executes the *setupRenting* function of the smart contract to set the rental period, rental price, deposit, and user (i.e., tenant). As a result, the ERC-4519 NFT modifies accordingly the *rentalTime*, *rentalPrice*, and *deposit* attributes; sends the *ReadingsRequest* notification to request the meter readings; and executes the *setUser* function (which sets the *state* attribute as “*waitingForUser*” and the *user* attribute as the tenant’s blockchain address, and sends the *UserAssigned* notification so that the smart home gateway knows that a new user has been assigned and is waiting to be engaged).The smart home gateway executes the *sendMeterReadings* function, which sets the water, electricity, gas, and diesel meter readings (0 if the service is not required). As a result, the ERC-4519 NFT updates the *Expenses_Meters* struct attribute with the values of the *waterMeter*, *dieselMeter*, *gasMeter,* and *electricityMeter*; updates the *timestamp* attribute; and sends the *ReadyHome* notification to let the tenant know that the smart home is ready to be rented.If the tenant agrees with the setup, the tenant generates some public authentication information, pays the rental price and the deposit, and executes the *startUserEngagement* function using the public authentication information. This function checks whether the tenant has paid the rental price and the deposit, if the meter readings have been updated, and if the timestamp is correct. It also initializes the mutual authentication process between the tenant and the smart home gateway, and saves the public authentication information in the ERC-4519 NFT, requesting engagement with the smart home gateway.The smart home gateway also generates some authentication information and uses it to execute the *userEngagement* function. The ERC-4519 NFT checks whether the stored public authentication information matches the authentication information received from the gateway. If they match, the ERC-4519 updates the *timestamp* attribute, sets the *state* attribute to “*engagedWithUser*”, pays the rental price to the landlord, and sends the *UserEngaged* notification to the smart home gateway and the tenant.

The setup process is illustrated in Figure 2.

#### 4.2.2. Rental Renewal

If the same tenant and smart home are available, the rental renewal process is run to change the rental price and/or period. Then, the following steps are carried out:The landlord executes the *renewRenting* function with the new rental price and period. As a result, the ERC-4519 NFT modifies accordingly the associated attributes and sends the *ReadingsRequest* notification to the smart home gateway.The smart home gateway executes the *updateMeterReadings* function. As a result, the ERC-4519 NFT updates the *Expenses_Meters* and *timestamp* attributes, calculates the expenses by executing the *calculateExpenses* function, updates the rental price with the calculated expenses, and sends the *ReadyHome* notification to inform the tenant that the smart home is ready for renewal.The tenant pays the rental price by executing the *payRenewal* function. As a result, the ERC-4519 NFT pays the increment to the landlord or the decrement to the tenant, updates the rental period, and sends the *RentingRenewed* notification to the smart home gateway.

This process is illustrated in Figure 3.

#### 4.2.3. Rental Termination

When the rental period expires, the following steps are carried out:The smart home gateway executes the *contractTermination* function and sends the meter readings.The ERC-4519 NFT updates the *timestamp* attribute and calculates the tenant’s expenses by executing the *calculateExpenses* function.If the expenses are greater than the deposit, the ERC-4519 NFT pays the deposit to the landlord, executes the *endTenancy* function to set all the attribute values to 0 and change its state to “*engagedWithOwner*”, and sends the *EmptyHome* event to the landlord and the smart home gateway.

This process is illustrated in Figure 4.

If the expenses are smaller than the deposit, the *deposit* attribute is updated to reflect the difference between the previous value and the expenses, and the ERC-4519 NFT pays the expenses to the landlord. The payment of the deposit to the tenant depends on whether there are home issues. If there are no home issues, the ERC-4519 NFT returns the updated deposit to the tenant, executes the *endTenancy* function to set all attribute values to 0 and change its state to “*engagedWithOwner*”, and sends the *EmptyHome* event to the landlord and the smart home gateway. This process is illustrated in Figure 5.

If there are home issues, the ERC-4519 NFT sends the *HomeIssues* notification to the landlord. Then, the landlord executes the *fixIssues* function, which updates the *homeIssues* attribute and sets the cost of the issues (which saturates to the deposit value). The ERC-4519 NFT pays the cost of the issues to the landlord, subtracts this cost from the deposit, sends the result to the tenant if it is greater than 0, executes the *endTenancy* function to set all attribute values to 0 and change its state to “*engagedWithOwner*”, and sends the *EmptyHome* event to the landlord and the smart home gateway. This process is illustrated in Figure 6.

If the tenant wants to leave the smart home before the rental period expires, the following steps are carried out:The tenant executes the *contractCancellation* function.The ERC-4519 NFT sends the *ReadingsRequest* event to the smart home gateway in order to receive the updates on the meter readings for the contract cancellation.The smart home gateway executes the *contractTermination* function, which sends the meter readings, and the rental termination continues as described above.

This process is illustrated in Figure 7.

## 5. Proof-of-Concept Realization and Experimental Results

The scheme of the proposed system for applying the ERC-4519 to the rental of smart homes is shown in Figure 8. The proof of concept for the application is based on the development of (a) the crypto wallets, (b) the ERC-4519 NFT smart contract, (c) decentralized applications (dApps) for the landlord and the tenant, and (d) the firmware for the smart home gateway. Experimental results were obtained in terms of the execution times for completing transactions and the costs of smart contract functions.

### 5.1. Development of the Crypto Wallets

MetaMask was employed as a crypto wallet for storing and managing the blockchain account keys, broadcasting transactions, sending and receiving cryptocurrencies and tokens, and securely connecting to the decentralized applications. All roles (tenant, landlord, smart home gateway, and ERC-4519 NFT) integrated MetaMask and the Alphawallet web3e library for Ethereum connection. Blockchain nodes were managed through Infura. The transaction messages exchanged were created using JSON (JavaScript Object Notation).

### 5.2. Development of the ERC-4519 NFT Smart Contract

The smart contract for renting smart homes described in the previous section was coded in Solidity. The Remix Integrated Development Environment was employed to develop, compile, and deploy the smart contract.

The smart contract was deployed in the Goerli Testnet, which is a testing network for Ethereum that allows experimenting without disturbing the main blockchain or losing real money. The smart contract has the address = 0xF83e8dCB6DC655E477d701d7A06ad59716dDC4b6.

### 5.3. Development of Landlord and Tenant dApps

Decentralized applications (dApps) were developed to allow the landlord and the tenant to interact with the smart home gateway. The landlord dApp allows checking if the smart home gateway is engaged with the landlord through the associated blockchain addresses and the *tokenId*. The landlord can initiate the engagement process using the dApp to set the ERC-4519 NFT state as “*engagedWithOwner*”. Similarly, the tenant dApp allows checking if the smart home gateway is engaged with the tenant and allows the tenant to execute the engagement process to set the ERC-4519 state as “*engagedWithUser*”. The dApps were developed using HTML and JavaScript with jQuery for web interfaces.

A middleware was required to connect the dApps with the smart home gateway. In this work, the middleware was developed using nodeJS. The dApps were connected to the middleware using WebSockets, and the smart home gateway was connected to the middleware using a UART connection. Hence, the tenant can check the authenticity of the smart home, preventing fraud on the part of the landlord, and the smart home can check the authenticity of its tenant, preventing unauthorized use by squatters.

### 5.4. Development of the Smart Home Gateway Firmware

The smart home gateway considered in this work was based on a Pycom Wipy 3.0 board with an ESP32 microcontroller. The firmware was developed to consist of two phases: registration and operation. During the registration phase, the smart home gateway generates a 32-byte secret key, uses the elliptic curve secpt256k1 to obtain the 64-byte public key (as established in Ethereum), and generates the blockchain address as the rightmost 20 bytes of the Keccak hash of the public key. That address ties the smart home gateway to its associated ERC-4519 NFT.

During the operation phase, the smart home gateway uses its blockchain address to complete blockchain transactions by preparing messages with a JSON structure, signing them, and transferring them to Infura.

The firmware was coded using Arduino and integrated into the platform.io as an extension of Microsoft Visual Studio Code. Trezor elliptic curves and the SHA3 Keccak hash from Alphawallet Web3e library were employed for the generation of the blockchain address.

### 5.5. Experimental Results

Table 5 shows the execution times of the operations carried out by the smart home gateway to complete a transaction. Execution times were obtained for a message with a size of 32 bytes. These times are insignificant compared to the time required by the blockchain to compute functions.

Table 6 shows the transaction costs of the functions associated with the ERC-4519 NFT smart contract. The functions *endTenancy* and *calculateExpenses* are not included because these functions are executed through other functions. The gas price considered was 17 gwei (1 gwei = ETH 0.000000001) and the price of an Ether (ETH) was USD 1887.51. These values were obtained on 30 June 2023 at 12.35 CET. The function with the highest cost is the smart contract deployment. However, it is only performed once. Similarly, the function *createToken* is only executed once for each smart home gateway. When comparing the costs with those of a real estate agency, it can be seen that the costs are much lower, maintaining the reliability of the tenant’s payment.

### 5.6. Comparison with State-of-the-Art Works

Table 7 compares this work with others reported in the literature. It can be seen that this work is the only one that considers the smart home as a non-fungible token and allows it to participate actively in the blockchain. In [1,2,3], smart contracts are used to replace physical contracts and eliminate intermediaries. In [6], the smart home is recognized on the blockchain as an entity that does not directly interact with the blockchain and authenticates the tenant and landlord off-chain. In our work, the smart home is a token that performs on-chain authentication. Since the home is smart and has the ability to generate its own blockchain account, it can interact with the blockchain to authenticate its tenant and owner, and similarly, both the tenant and owner can authenticate the smart home. Concerning transaction costs, only the work in [1] provides the total cost of renting. This is shown in Table 7, expressed in Ethers. On the other hand, the total cost shown for this work considers the renting of a smart home already engaged with its owner (i.e., taking into account the transaction costs shown in Table 6 for the functions *setupRenting*, *setUser*, *startUserEngagement*, *userEngagement*, *setExpensesPrices*, *sendMeterReadings*, and *contractTermination*). Concerning execution times, only the work in [3] provides the generation time of the key needed for the authentication process, which is performed off-chain between the tenant and the landlord. This is shown in Table 7, expressed in seconds. On the other hand, the generation time of the key shown for this work considers the time needed by the smart home to generate the public key that allows it to authenticate with the landlord and the tenant, which is also shown in the first row of Table 5.

## 6. Conclusions

This paper explains how the ERC-4519 NFT can be used to design a smart contract for home rental in Ethereum. All the attributes, events, and functions of the rental process are described. Among the latter, the rental setup, renewal, and termination are detailed. The smart home is tied to the NFT through its own blockchain account, and the NFT includes the operating modes of the smart home gateway. Communications are secure because mutual authentication is implemented between the landlord/tenant and the smart home.

The smart contract was programmed in Solidity and deployed on the Goerli Testnet for Ethereum. The transaction costs of the relevant functions of the smart contract are low, much lower than the costs of real estate agencies. The gateway of the smart home was implemented in a Pycom Wipy 3.0 board with an ESP32 microcontroller. It takes a few tens of milliseconds to complete a transaction, which is quite affordable for common applications and does not affect the main functionality of the gateway.

The contributions of the proposal based on ERC-4519 NFTs to the smart home rental market can be evaluated from several perspectives. The smart contract information is stored on the blockchain, thus allowing tamper-proofing and traceability for the preservation of the rental contracts. However, this information may not have legal recognition depending on the country. Regarding fungibility, the proposal ties the smart home to a non-fungible token but provides a mechanism to fix the fungible cost of any possible damage caused by the tenant (the smart home notifies the landlord of any issues, and the landlord determines the cost and notifies the NFT to reduce the deposit). Regarding price volatility and valuation, rental prices can be established during the rental setup and renewal processes. In terms of flexibility, the proposal is decentralized by eliminating intermediaries between the landlord and the tenant. Agreements and payments are made online and automatically, reducing time and management resources.

In future work, we plan to explore the application of ERC-4519 NFTs in other domains. We will consider applications that require users but do not require a link between the physical asset and the NFT, such as lending books, unique game cards, or clothing. We will also consider applications that require a link between the physical asset and the NFT but do not require users, such as the control of dangerous objects like firearms and luxury objects like cars.

## Figures and Tables

**Figure 1 sensors-23-07101-f001:**
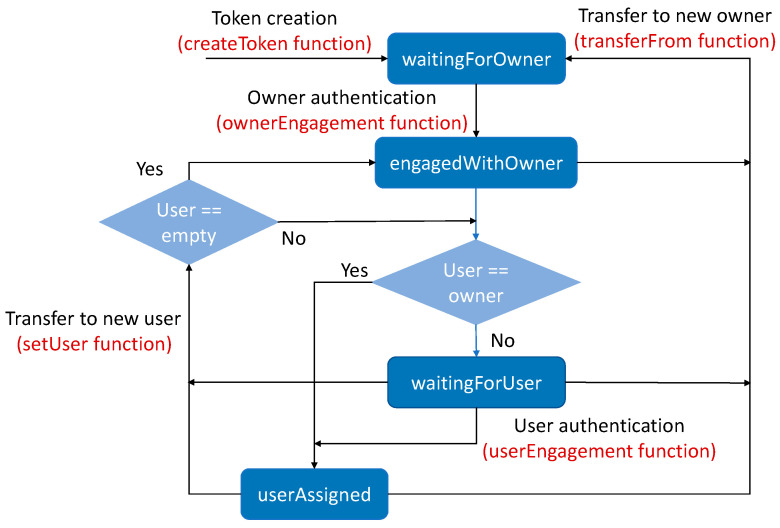
Flowchart of the ERC-4519 NFT states, with the *asset* and *user* addresses defined.

**Figure 2 sensors-23-07101-f002:**
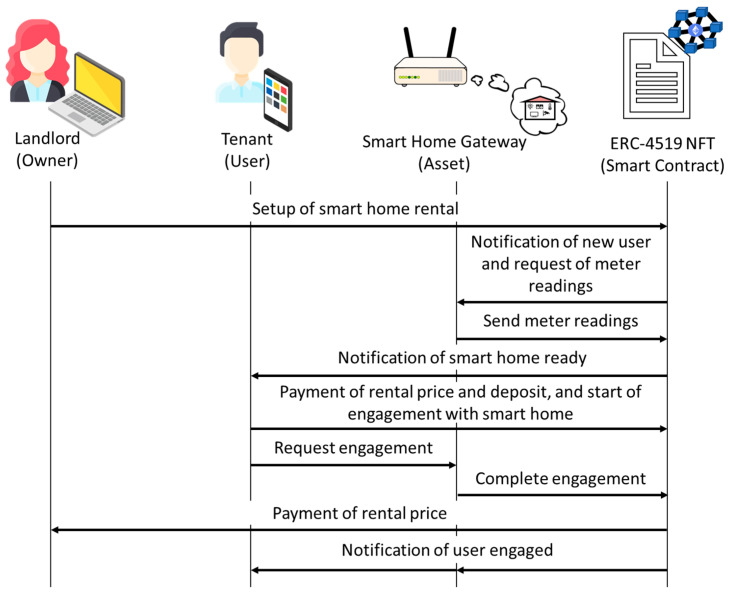
Rental setup.

**Figure 3 sensors-23-07101-f003:**
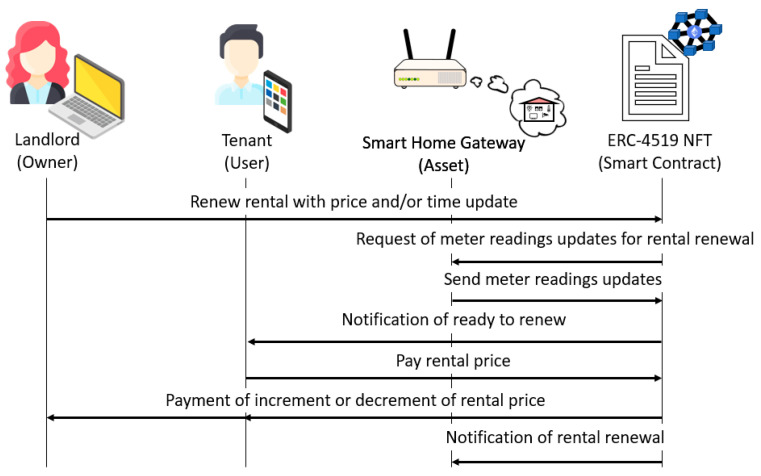
Rental renewal.

**Figure 4 sensors-23-07101-f004:**
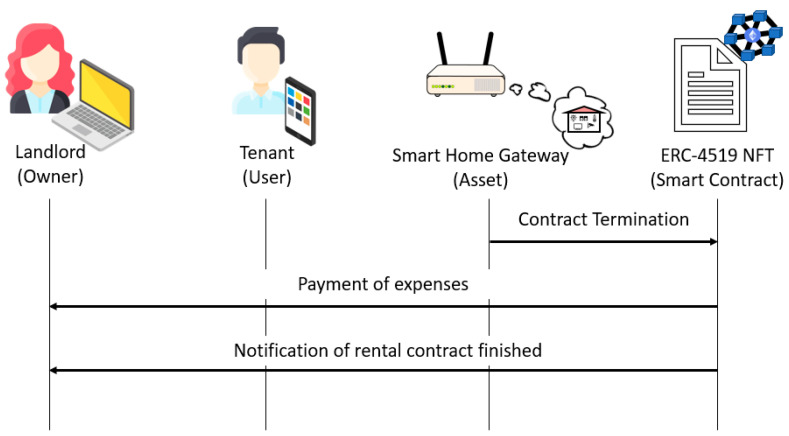
Rental termination if expenses are greater than the deposit.

**Figure 5 sensors-23-07101-f005:**
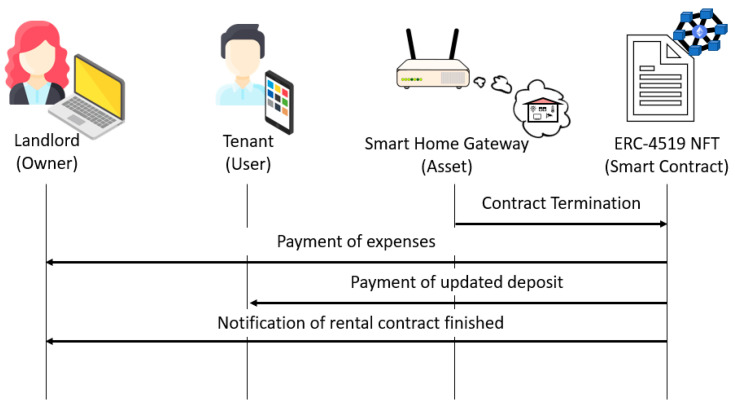
Rental termination if expenses are smaller than the deposit and there are no home issues.

**Figure 6 sensors-23-07101-f006:**
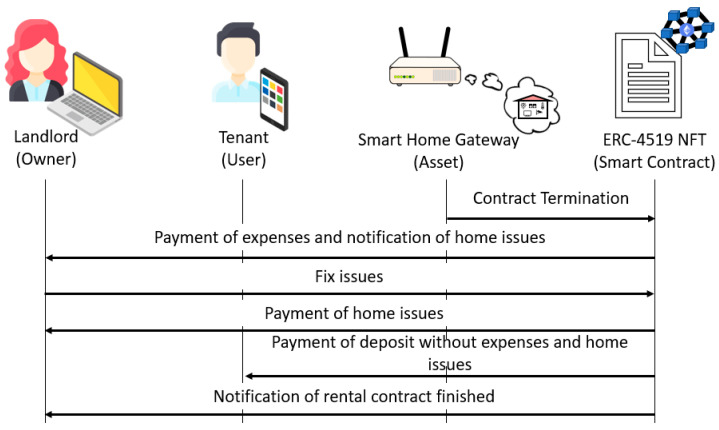
Rental termination if the deposit covers expenses and there are home issues.

**Figure 7 sensors-23-07101-f007:**
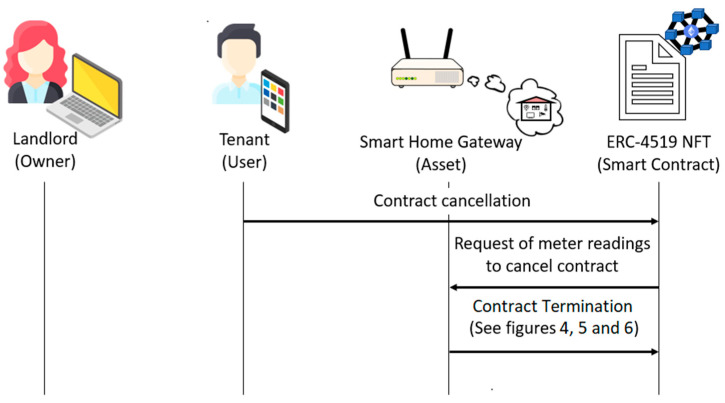
Rental termination by contract cancellation.

**Figure 8 sensors-23-07101-f008:**
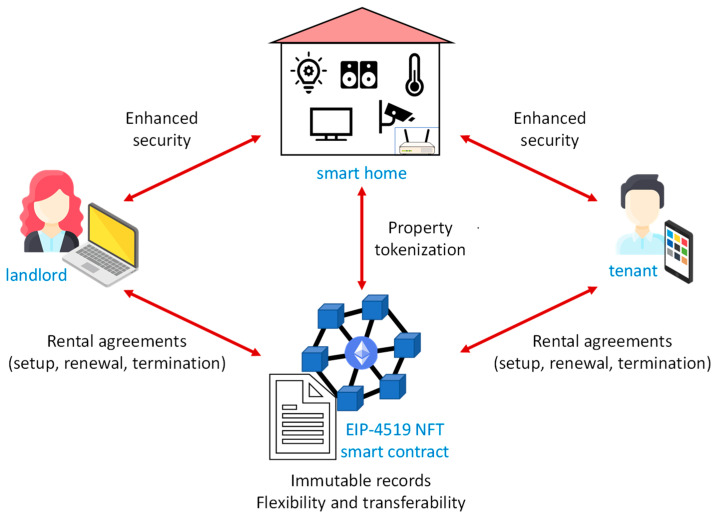
Scheme of the proposed system.

**Table 1 sensors-23-07101-t001:** Summary of related works concerning blockchains and smart homes.

Refs.	Focus on Rental	Focus on Property Transference	IoT-Based Access Control	Avoidance of Intermediaries	Home Tokenization
[1]	Yes	No	No	Yes	No
[2]	Yes	No	No	Yes	No
[3]	Yes	No	No	No	No
[4]	No	Yes	Yes	Yes	No
[6]	Yes	No	Yes	Yes	No
[9]	No	Yes	No	Yes	Yes (Fungible)
[15]	No	No	Yes	Yes	No
[16]	No	Yes	No	Yes	Yes (Non-Fungible)

**Table 2 sensors-23-07101-t002:** Comparison of ERCs for NFTs.

Refs.	ERC Name	NFT with Users	NFT Tied to Asset	NFT with States	NFT Enabling Secure Communication	Status
[10]	ERC-721	No	No	No	No	Final
[14]	ERC-777	No	No	No	No	Final
[17]	ERC-1155	No	No	No	No	Final
[18]	ERC-4494	Yes	No	No	No	Draft
[19]	ERC-5334	Yes	No	No	No	Draft
[20]	ERC-4907	Yes	No	No	No	Final
[21]	ERC-5187	Yes	No	No	No	Draft
[22]	ERC-5501	Yes	No	No	No	Draft
[23]	ERC-2615	Yes	No	No	No	Stagnant
[11]	ERC-4519	Yes	Yes	Yes	Yes	Final

**Table 3 sensors-23-07101-t003:** Attributes considered in smart homes represented by ERC-4519 NFTs.

Type	Attributes	Type	Attributes
uint256	* tokenId *	uint256	*deposit*
address	* owner *	uint256	*rentalPrice*
address	* user *	uint256	*waterMeter/waterPrice*
address	* asset *	uint256	*dieselMeter/dieselPrice*
enum States	* state *	uint256	*gasMeter/gasPrice*
uint256	* hashK_OA *	uint256	*electricityMeter/electricityPrice*
uint256	* hashK_UA *	uint256	*rentalTime*
uint256	* dataEngagement *	uint256	*homeIssues*
uint256	* timestamp *	struct	*Expenses_Meters*
uint256	* timeout *	struct	*Expenses_Prices*

**Table 4 sensors-23-07101-t004:** Functions of the smart contract.

Function Name	Input Attributes
* createToken *	* asset, owner *
* transferFrom *	* tokenId, addressFrom, addressTo *
* setUser *	* tokenId, user *
* startOwnerEngagement *	* tokenId, hashK_OA, dataEngagement *
* ownerEngagement *	* hashK_OA *
* startUserEngagement *	* tokenId, hashK_UA, dataEngagement *
* userEngagement *	* hashK_UA *
*setupRenting*	*tokenId, user, rentalTime, rentalPrice, deposit*
*sendMeterReadings*	*waterMeter, electricityMeter, gasMeter, dieselMeter*
*renewRenting*	*tokenId, newRentalTime, newRentalPrice*
*updateMeterReadings*	*waterMeter, electricityMeter, gasMeter, dieselMeter*
*payRenewal*	*tokenId*
*contractTerminator*	*waterMeter, electricityMeter, gasMeter, dieselMeter*
*contractCancellation*	*tokenId*
*newHomeIssue*	*-*
*fixIssues*	*tokenId, issueCost*
*setExpensesPrices*	*waterPrice, electricityPrice, gasPrice, dieselPrice*
*getExpensesPrices*	*tokenId*
*endTenancy*	*tokenId*
*calculateExpenses*	*tokenId, waterMeter, electricityMeter, gasMeter, dieselMeter*

**Table 5 sensors-23-07101-t005:** Execution times of operations carried out by the smart home gateway.

Operation	Execution Time (ms)
Generation of the 64-byte public key (secp256k1)	21.15
Generation of the 20-byte blockchain address (Keccak256)	0.45
Transaction message preparation (JSON and signature)	26.10
Transfer to Infura	2.90
Transaction completion	50.60

**Table 6 sensors-23-07101-t006:** Transaction costs of the functions of the smart contract.

Function	Transaction Cost (Gas)	Transaction Cost (ETH)	Transaction Cost (USD)
*Deployment*	5,173,251	0.0879453	166.00
*createToken*	211,963	0.0036034	6.80
*startOwnerEngagement*	76,687	0.0013037	2.46
*ownerEngagement*	58,500	0.0009945	1.88
*setupRenting*	144,608	0.0024583	4.64
*sendMeterReadings*	131,562	0.0022366	4.22
*startUserEngagement*	86,856	0.0014766	2.79
*userEngagement*	53,683	0.0009126	1.72
*setExpensesPrices*	51,129	0.0008692	1.64
*renewRenting*	99,888	0.0016981	3.21
*updateMeterReadings*	121,306	0.0020622	3.89
*payRenewal*	63,155	0.0010736	2.03
*newHomeIssue*	60,468	0.0010280	1.94
*contractCancellation*	50,109	0.0008519	1.61
*contractTermination*	81,279	0.0013817	2.61
*fixIssues*	72,534	0.0012331	2.33
*setUser*	51,734	0.0008795	1.66
*transferFrom*	66,300	0.0011271	2.13

**Table 7 sensors-23-07101-t007:** Comparison of our work with state-of-the-art works.

Considerations	[1]	[2]	[3]	[6]	This Work
Smart home tokenization	No	No	No	No	Yes
On-chain authentication between home and tenant	No	No	No	No	Yes
Transaction costs per rental (ETH)	0.01095	-	-	-	0.01021
Generation time of key for authentication (s)	-	-	117.3	-	0.021

## Data Availability

Not applicable.

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
