# Peer review of "Non-Fungible Tokens Based on ERC-4519 for the Rental of Smart Homes"

_sensors, 2023, doi:10.3390/s23167101_

Round 1

Reviewer 1 Report

Dear authors, 

I believe you need to improve your methodology in order to increase the quality of the paper. Please be specific in your choice - qualitative, review, empirical, etc and revise accordingly. Once you have revised this, please check the rest of the paper for concordance. 

Minor revisions of spelling. 

Reviewer 2 Report

A housing rental scheme based on virtual currency is proposed in the manuscript. To be honest, the manuscript is more like a technical implementation report. Whether it is ERC-4519 or other potential future standards, its design purpose is for trading. The manuscript specifically applies it to a certain scenario (rental housing). As a journal paper, I think the content is appropriate and the protocol description is clear and complete. 

Reviewer 3 Report

The manuscript demonstrates strong writing skills, and the authors' contribution appears promising. However, I have a suggestion regarding the proposed model. In order to validate their approach, it would be beneficial for the authors to include either a simulation or a mathematical model that supports their claims.

To provide further support, I recommend considering the utilization of either SpAN AVISPA or BAN Logic. These tools can aid in the validation process and enhance the credibility of the proposed model. Incorporating one of these tools would allow the authors to demonstrate the effectiveness and robustness of their model in a more concrete and rigorous manner.

Overall, by including a simulation or math model, along with the suggested tools, the authors can strengthen the validity of their proposed model and provide additional evidence to support their claims

Reviewer 4 Report

Title: “Non-Fungible Tokens Based on ERC-4519 for the Rental of Smart Homes”

The concept of Non-Fungible Tokens (NFTs) is based on the ERC-4519 standard, specifically designed for the rental of smart homes. NFTs have gained significant attention in recent years as unique digital assets that can represent ownership or proof of authenticity for a wide range of items, including artwork, collectibles, and virtual real estate. Now, with the emergence of ERC-4519, one can explore their potential application in the realm of smart home rentals as described in the manuscript.

Moreover, ERC-4519 is a protocol built on the Ethereum blockchain, which provides a framework for the creation, ownership, and transfer of NFTs specifically tailored for the rental market of smart homes. By leveraging the benefits of blockchain technology and smart contracts, ERC-4519 enables a secure and transparent system for managing smart home rentals.

Even though the article presents some useful experiments performed for target approaches, some aspects should be improved for possible publication and for a better understanding. So, the author(s) are advised to perform the following changes to be considered for possible publication.

1.      Why a phrase “a few American dollars” is specifically used in the manuscript? Research is should not be specific to a currency name, this may be written in a generalized way.

2.      The introduction section of the manuscript lacks consistency and coherency among multiple sentences. This should be corrected accordingly.

3.      A theoretical comparative analysis table should be added to the section to show the existing frameworks in terms of  their strengths, and their weaknesses.

4.      A graphical representation of ERC 4519 should be added to section “3. ERC-4519: Non-fungible Tokens Tied to Physical Assets”. For ease of understanding and clarity.

5.      The algorithm given in “Table 2. Functions of the smart contract” should be in proper algorithmic format with indentation.

6.      A block diagram of the proposed system should be added to the manuscript to show the overall working of the proof of concept. In block diagram there should be labels of the following:

a.      Property Tokenization

b.      Rental Agreements

c.      Immutable Records

d.      Flexibility and Transferability

e.      Enhanced Security

7.      The “State-Of-Art Comparison”  presented in section 5.6 does not show any evidence of such comparison to show that why the proposed work is optimal among them. So, there should be comparative analysis based on the measurement parameters i.e., execution time and transaction cost.

8.      In the abstract it is claimed that proposed work will prevent fraud or squatters etc. but there are no such parameters that have been measured or discussed to justify the claim.

9.      The conclusion of the manuscript should be rewritten to concisely/briefly justify what were the issues and what has been achieved. Moreover, there should be some future research direction in the conclusion section to attract the readers.

10.   Moreover, it is important to consider the limitations and evaluate the suitability of ERC-4519 NFTs for rental agreements based on the specific requirements and circumstances of the smart home rental market.:

a.      Lack of legal recognition

b.      Lack of fungibility

c.      Price volatility and valuation

d.      Limited flexibility

Minor changes are required

Round 2

Reviewer 1 Report

Dear authors, I do appreciate the efforts you have made to improve the work. Thanks 

Reviewer 3 Report

the manuscript has been Accepted in the current format the authors have been taken into consideration all comments

Reviewer 4 Report

Title: “Non-Fungible Tokens Based on ERC-4519 for the Rental of Smart Homes”

I reviewed the revised version of the above-entitled manuscript and found out that the authors satisfactorily incorporated all the required changes suggested in the first round of review. Therefore, I hereby accept the manuscript for publication.